# Structural basis for catalysis by human lipoyl synthase

Olga A. Esakova [1,2] ✉, Douglas M. Warui[1,2], Syam Sundar Neti [1,2],
John N. Alumasa[1,6] & Squire J. Booker [1,2,3,4,5] ✉

Lipoic acid is an essential cofactor in five mitochondrial multiprotein complexes. In each complex, it is tethered in an amide linkage to the side chain of a conserved lysyl residue on a lipoyl carrier protein or lipoyl domain to afford the lipoyl cofactor. Lipoyl synthase catalyzes the last step in the biosynthesis of the lipoyl cofactor, the addition of two sulfur atoms to carbons 6 and 8 of an octanoyllysyl residue of the H protein, the lipoyl carrier protein of the glycine cleavage system. Lipoyl synthase, a member of the radical S-adenosylmethionine superfamily, contains two $[Fe_4S_4]$ clusters, one of which is sacrificed during catalysis to supply the appended sulfur atoms. Herein, we use X-ray crystallography to characterize several stages in lipoyl synthase catalysis and present a structure of an intermediate wherein the enzyme is cross-linked to the H protein substrate through a 6-mercaptooctanoyl ligand to a $[Fe_3S_4]$ cluster.

Lipoic acid (LA) is an essential sulfur-containing and redox-active molecule used as a cofactor in several multiprotein complexes (Fig. 1)[1,2]. In its cofactor form, it is covalently attached to a specific lysyl residue on lipoyl carrier proteins (LCPs) or lipoyl domains (LDs) in these complexes[1–3]. In humans, five mitochondrial multiprotein complexes employ LA as a cofactor. Four among these are the 2-oxoacid dehydrogenase complexes, which include the α-ketoglutarate dehydrogenase (KDC), pyruvate dehydrogenase (PDC), α-ketoadipate dehydrogenase (KAC), and branched-chain oxo-acid dehydrogenase complexes (BCODC). All of these complexes are composed of similar $E_1$, $E_2$, and $E_3$ subunits, wherein the $E_2$ subunits are acyltransferases that contain LDs[2,4]. The fifth complex is the glycine cleavage system (GCS), which couples the decarboxylation of glycine with the production of methylenetetrahydrofolate and ammonia. It comprises four subunits: P, T, L, and H proteins[5]. LA is covalently attached to the ε-amino group of lysine 107 of the human H-protein ($H_{pro}$)[6].

The biosynthetic pathway for the lipoyl cofactor (LipCo) was initially established for *Escherichia coli*[7]. This organism uses two pathways to generate the LipCo[8]. In the exogenous or salvage pathway, *E. coli* employs a bifunctional lipoate protein ligase A (LplA) to adenylate free LA and transfer it to the organism's three LCPs[9]. By contrast, the endogenous pathway is an offshoot of type 2 fatty acid biosynthesis. In this pathway, an *n*-octanoyl chain is constructed on an acyl carrier protein (ACP) and then transferred to the three LCPs in the organism by octanoyl protein ligase B (LipB)[10–12]. Lipoyl synthase, LipA in *E. coli* and most other bacteria, then attaches two sulfur atoms to the octanoyl chains of the LCPs to complete the pathway[13,14]. Many other bacteria use similar strategies for constructing the LipCo, although both pathways are not always present[15]. Moreover, some bacteria and other organisms contain important modifications to the strategy used by *E. coli*. For example, the LipCo is constructed only on the $H_{pro}$ in humans, yeast, and some bacteria. A separate amido-transferase then distributes the lipoyl group to the $E_2$ subunits of the 2-oxoacid dehydrogenase complexes[16–18]. In humans, LipCo biosynthesis occurs in the mitochondria and begins with the transfer of an octanoyl moiety from the mitochondrial acyl-carrier protein (ACP1) to Lys107 of the $H_{pro}$[19]. This transfer is catalyzed by octanoyltransferase 2 (LIPT2) and forms octanoyl-$H_{pro}$ (OCT-$H_{pro}$)[20,21]. Subsequently, LIAS attaches two sulfur atoms at C6 and C8 of the octanoyl moiety[22–24].

[1]Department of Chemistry, The Pennsylvania State University, University Park, PA, USA. [2]Department of Chemistry, School of Arts and Sciences, University of Pennsylvania, Philadelphia, PA, USA. [3]Department of Biochemistry & Molecular Biology, The Pennsylvania State University, University Park, PA, USA. [4]Howard Hughes Medical Institute, Chevy Chase, MD, USA. [5]Department of Biochemistry and Biophysics, Perelman School of Medicine at the University of Pennsylvania, Philadelphia, PA, USA. [6]Present address: Department of Chemistry and Biochemistry, Miami University, Oxford, OH, USA.
✉e-mail: oae3@sas.upenn.edu; sjbooker@mac.com

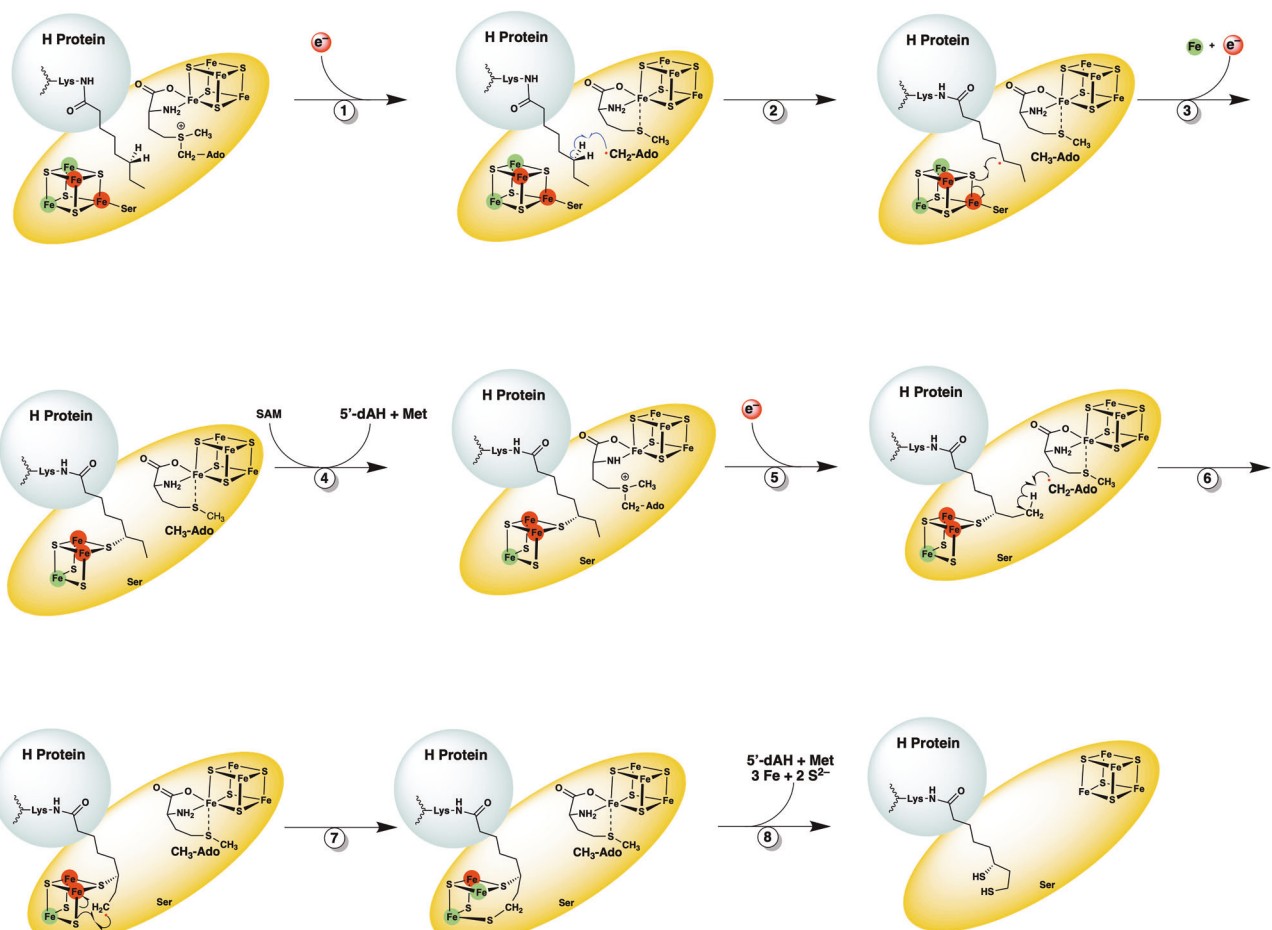

**Fig. 1 | The LIAS-catalyzed reaction.** The LIAS-generated lipoyl cofactor is shown in oxidized and 2-electron reduced forms.

**Fig. 2 | Current LIAS reaction mechanism.** Reduction of the RS iron/sulfur ([Fe₄S₄]RS) cluster[1] results in cleavage of SAM to generate a 5'-dA•, which abstracts the C6 *pro-R* hydrogen from the octanoyllysyl substrate[2]. The resulting C6 substrate radical attacks a bridging μ-sulfido ion of the auxiliary iron/sulfur ([Fe₄S₄]AUX) cluster with dissociation of the Ser352 ligand and concomitant loss of Fe²⁺ (Fe²⁺ shown in green; Fe³⁺ shown in red) and an electron[3] to yield a [Fe₃S₃:1 mercaptooctanoyllysyl]⁰ intermediate adduct. In the second half-reaction, a second SAM molecule binds[4] and is cleaved to a second 5'-dA•[5], which abstracts a C8 hydrogen atom[6]. The resulting C8 radical attacks a second bridging sulfido ion of the auxiliary cluster with concomitant reduction of a Fe³⁺ to Fe²⁺[7]. The resulting unstable species breaks apart, releasing 3Fe and 2S²⁻, as well as the lipoyllysyl group, upon the addition of two protons[8].

LIAS belongs to the radical *S*-adenosylmethionine (SAM) superfamily of enzymes and harbors two [Fe₄S₄] clusters[14,25–28]. One cluster, the radical SAM (RS) cluster or [Fe₄S₄]RS, is coordinated by three cysteinyl residues in a conserved CX3CX2C motif, leaving one Fe ion able to be coordinated by the amino and carboxylate groups of SAM[25,29]. The second cluster, the auxiliary cluster or [Fe₄S₄]AUX, is coordinated by cysteinyl residues in a CX4CX5C motif and a seryl residue in a C-terminal RSSY motif[29,30]. The [Fe₄S₄]AUX cluster is sacrificed during catalysis to provide the appended sulfur atoms[29,31,32]. In the LIAS

reaction, which proceeds through two half-reactions, two molecules of SAM are reductively cleaved to generate two molecules of methionine (Met) and two 5'-deoxyadenosyl 5'-radicals (5'-dA•). In the first half-reaction, a 5'-dA• abstracts a hydrogen atom (H•) from C6 of OCT-Hpro, generating a C6-centered radical that attacks a bridging μ-sulfido ion of the auxiliary cluster, forming a covalently bound intermediate[29,31,33,34]. In the second half-reaction, a second 5'-dA• abstracts a H• from C8 of the covalently bound intermediate, generating a C8-centered radical that attacks a second μ-sulfido ion of the auxiliary cluster, producing

the lipoylated H-protein (Lip-H$_{pro}$) (Fig. 2)[32,33]. Unlike in *E. coli*, the LIAS reaction in humans is believed to occur only on the H$_{pro}$. Another enzyme, lipoyl transferase 1 (LIPT1), transfers the lipoyl group from Lip-H$_{pro}$ to the E$_2$ subunits of the 2-oxoacid dehydrogenase complexes[7,17,35–38]. Because LIAS degrades its [Fe$_4$S$_4$]$_{AUX}$ cluster during catalysis, it can catalyze no more than one turnover in vitro[23,24]. Recent studies have shown that the iron-sulfur (FeS) carrier protein NFU1 can reconstitute the [Fe$_4$S$_4$]$_{RS}$ cluster[22] and also the [Fe$_4$S$_4$]$_{AUX}$ cluster after each turnover, rendering LIAS catalytic[24].

An alternative pathway for generating the LipCo in several archaeal organisms has recently been discovered and is distinct from the pathways in eukaryotes and bacteria[39]. In this pathway, the classical lipoyl synthase is replaced by two RS proteins, LipS1 and LipS2. These two proteins share poor sequence identity with classical LSs[39]; however, they both contain [Fe$_4$S$_4$]$_{RS}$ and [Fe$_4$S$_4$]$_{AUX}$ clusters[40]. In contrast to the order of sulfur attachment in classical LSs, LipS2 first appends a sulfur atom at C8 of an octanoyllysyl-containing substrate, while LipS1 appends a sulfur atom at C6[39,40].

Structures of bacterial lipoyl synthases from *Mycobacterium tuberculosis* and *Thermosynechoccus elongatus* have been reported[29,30]. The *T. elongatus* LipA structure was not determined with a biologically relevant ligand. By contrast, a peptide substrate containing an octanoyllysyl moiety was used to capture an intermediate in the *M. tuberculosis* LipA reaction. This intermediate contained a bond between C6 of the substrate and one of the sulfide ions of the [Fe$_3$S$_4$]$_{AUX}$ cluster. Also, Met was bound to the [Fe$_4$S$_4$]$_{RS}$ cluster, and the 5′ carbon of 5′-deoxyadenosine (5′-dAH) was 4.2 Å from C6 and 3.7 Å from C8 of the octanoyl chain[29]. This structure was consistent with H• abstraction occurring initially at C6 and the auxiliary cluster being degraded to supply the attached sulfur atom. However, the structure does not provide details about other aspects of the reaction, such as the state of the active site just before the reductive cleavage of SAM for C6 or C8 attachment. Moreover, the structure was determined using a peptide substrate rather than a physiological protein substrate.

In this work, we present detailed structural snapshots of several catalytic steps involved in LipCo formation, including the structure of LIAS in complex with its physiological substrate, H$_{pro}$. Structures include complexes with a noncovalently bound peptide and with the H$_{pro}$ in the presence of SAM or 5′-dAH + Met. These structures provide mechanistic insights into the reaction steps that occur before and after H• abstraction, as well as active site details that enable the protein to modify C6 or C8 selectively. Additionally, these structures provide details of the interface interactions between LIAS and H$_{pro}$. Lastly, our structural study enabled the mapping of pathogenic variants in human LIAS and H$_{pro}$, providing molecular insight into their functional consequences. The analysis reveals that specific disease-causing substitutions in LIAS, such as Glu159Lys, Asp215Glu, Met310Thr, and Arg249His, affect Fe/S cluster coordination, SAM binding, or the interaction with the H$_{pro}$.

## Results

### The overall structure of human LIAS
The structure of LIAS was determined to 1.54 Å resolution and reveals a monomeric enzyme in the asymmetric unit (Fig. 3a). The overall structure is similar to those of *M. tuberculosis* LipA (*Mt*LipA)[29] (PDB ID: 5EXJ, RMSD of 1.1 Å over 240 Cα-atoms) (Supplementary Fig. 1a) and *T. elongatus* LipA (*Te*LipA) (PDB ID: 4U0P, RMSD of 0.94 Å over 226 Cα-atoms) (Supplementary Fig. 1b)[30], consisting of an N-terminal domain, a central (βα)$_6$ partial TIM barrel RS domain, and a C-terminal α-helix. The structure shows the presence of both [Fe$_4$S$_4$] clusters in their intact states. Three Fe of the [Fe$_4$S$_4$]$_{RS}$ cluster are coordinated by cysteines 137, 141, and 144, while the fourth, unique Fe, which is typically ligated by SAM or Met, is bound by a molecule of dithiothreitol (DTT) (Fig. 3b). As in the *Mt*LipA and *Te*LipA structures, the RS domain of LIAS contains an extra β-strand extension, β7, that continues to the

protein's C-terminal α-helix. The N-terminal domain and the C-terminal α-helix provide ligands to the [Fe$_4$S$_4$]$_{AUX}$ cluster. The N-terminal domain (Fig. 3a, tan) contains three helices that harbor cysteines 106, 111, and 117 of the Cx$_4$Cx$_5$C motif, which ligate to three Fe of the [Fe$_4$S$_4$]$_{AUX}$ cluster. Ser352, in the conserved RS**S**$^{352}$Y motif just before the start of the C-terminal α-helix, coordinates the fourth Fe of the [Fe$_4$S$_4$]$_{AUX}$ cluster, which faces toward the active site of the protein (Fig. 3a, purple). The structures of LIAS, *Mt*LipA, and *Te*LipA exhibit some differences around the auxiliary clusters. Unlike in *Mt*LipA and *Te*LipA, the α-helix containing Cys117 in the LIAS structure is shorter (Supplementary Fig. 2a), and residues 118–126 of the loop are in close contact with the C-terminal α-helix of the protein (Supplementary Fig. 2b, purple), a feature found only in LIAS.

### Thiolation at C6 of the substrate octanoyl moiety
In the first step of LIAS catalysis, a 5′-dA• abstracts a H• from C6 of the OCT moiety of the substrate, generating a C6-centered substrate radical (Fig. 2, step 2). This substrate radical attacks a bridging µ-sulfido ion of the [Fe$_4$S$_4$]$_{AUX}$ cluster with concurrent or subsequent loss of an Fe$^{2+}$ ion and an electron to afford an [Fe$_3$S$_3$:1 mercaptooctanoyllysyl]$^0$ cluster[29,31] (Fig. 2, step 3). This cluster resembles an [Fe$_3$S$_4$]$^+$ cluster but has one of its sulfido ions covalently attached to C6 of the OCT moiety of the substrate. We captured the steps in catalysis involving the C6 H• abstraction by determining the structure of LIAS in the presence of 5′-dAH plus Met (5′-dAH+Met) and an eight amino acid (aa) peptide substrate mimic containing an *N*-octanoyllysyl group (OCT-8$_{mer}$). The structure was determined to 1.58 Å resolution, showing two subunits in an asymmetric unit (Supplementary Table 1). Only one of the subunits contains the substrate and 5′-dAH+Met in the active site, while the other contains only Met.

The structure shows the OCT-8$_{mer}$ substrate and 5′-dAH sandwiched between the two [Fe$_4$S$_4$] clusters, with the OCT moiety of the substrate positioned between 5′-dAH and the auxiliary cluster (Fig. 4a). On one side, C6 of the substrate is 3.0 Å away from C5′ of 5′-dAH, a distance suitable for H• abstraction. On the other side, C6 of the substrate is 3.2 Å away from the closest sulfur of the [Fe$_4$S$_4$]$_{AUX}$ cluster, an appropriate distance for the attack of the substrate radical. The binding of the OCT-8$_{mer}$ in the active site is predominantly through a network of van der Waals interactions and a key H-bond made between its carbonyl oxygen and the epsilon nitrogen of Arg350, a strictly conserved residue in the RSSY motif (Fig. 4b). In this structure, C8 of the substrate is pointed away from C5′ of 5′-dAH at a distance of 4.2 Å and is positioned between C$_\beta$ of Ser352 and the carbonyl carbon of Ser351 (Fig. 4a). This active site arrangement is consistent with sulfur attachment at C6 always preceding attachment at C8.

The binding of the substrate and 5′-dAH+Met leads to noticeable conformational changes, most significantly around the [Fe$_4$S$_4$]$_{AUX}$ cluster (Fig. 4c, green arrows). In addition, the loop between residues 131 and 156 containing the Cx$_3$Cx$_2$C motif exhibits a new conformation. Altogether, these changes decrease the distance between the two clusters from 13.7 Å to 11.8 Å. Moreover, the previously discussed loop (residues 118–126) that interacts with the C-terminal helix in the substrate-free structure is dissociated in the presence of the OCT-8$_{mer}$ and 5′-dAH+Met (Supplementary Fig. 2c). These changes lead to the closing of the substrate channel (Fig. 4e) and rearrangements in the electrostatic surface potential of LIAS (Fig. 4d, e).

### Thiolation at C8 of the [Fe$_3$S$_3$:1 mercaptooctanoyllysyl]$^0$ species
The structure of LIAS with the OCT-8$_{mer}$ and 5′-dAH+Met (Fig. 4a) represents a snapshot of the active site just after the reductive cleavage of SAM to a 5′-dA• and Met. In the next step, the 5′-dA• abstracts a H• from C6 of the substrate, generating a C6-centered radical that attacks a µ-sulfido ion of the auxiliary cluster, forming a covalently attached 6-thiolated intermediate (Fig. 2, steps 2 and 3). To mimic these steps in catalysis, we generated a complex between the OCT-H$_{pro}$

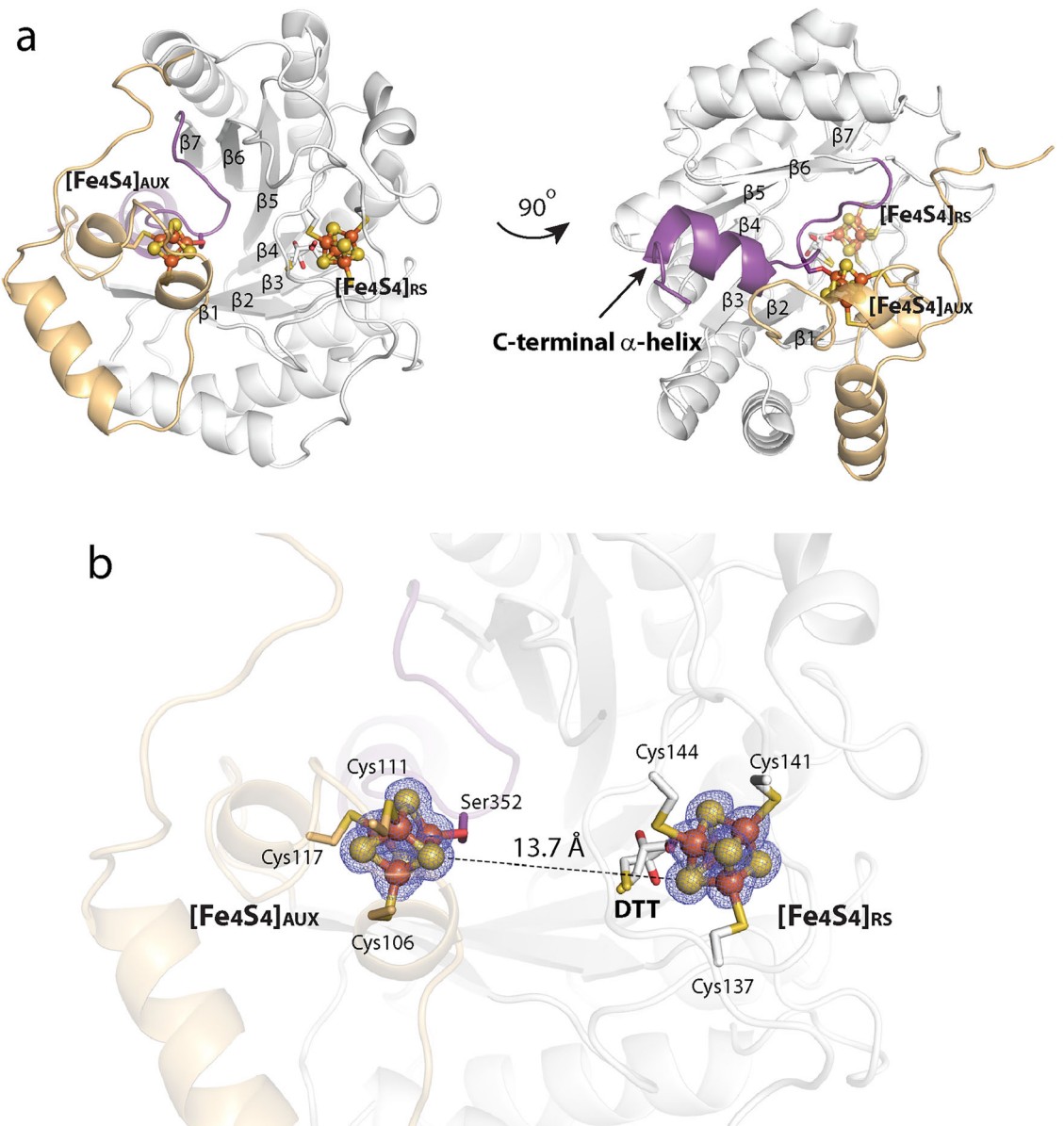

**Fig. 3 | The structure of human LIAS.** A cartoon representation of **a** the overall structure of LIAS; **b** the active site cavity of LIAS formed between two $[Fe_4S_4]$ clusters with dithiothreitol (DTT) (two conformations) bound to the RS iron-sulfur cluster ($[Fe_4S_4]_{RS}$) cluster. The mesh is contoured at 2.5 σ. LIAS color code: N-terminal domain (tan); RS domain (white); C-terminal α-helix (purple); iron–sulfur atoms of the auxiliary ($[Fe_4S_4]_{AUX}$) and $[Fe_4S_4]_{RS}$ clusters are represented as orange and yellow balls, respectively.

and LIAS where C6 of the OCT group is bound covalently to one of the sulfurs of the auxiliary cluster of LIAS, which we term the LIAS-6S-OCT-$H_{pro}$ intermediate. This complex was formed by incubating equimolar concentrations of LIAS, $H_{pro}$, and SAM in the presence of an excess of reductant. The complex was then purified and crystallized in the presence of SAM (Fig. 5a, b) or 5'-dAH+Met (Fig. 5c).

The structure of the LIAS-6S-OCT-$H_{pro}$ intermediate in the presence of SAM was determined to 2.58 Å resolution and contained three subunits in the asymmetric unit (Supplementary Table 1). However, the density for the $H_{pro}$ was discernible only in two subunits containing the complex. The third subunit showed the 6-mercaptooctanoyl intermediate in the active site; however, the $H_{pro}$ was disordered. In the LIAS-6S-OCT-$H_{pro}$ intermediate structure, the auxiliary cluster is in the $[Fe_3S_3;1$ mercaptooctanoyllysyl] state, and the conserved Ser352 of the RS**$S^{352}$**Y motif no longer coordinates the auxiliary cluster due to the loss of the unique Fe. The LIAS-6S-OCT-$H_{pro}$ intermediate is positioned

in the active site similarly to the OCT-$8_{mer}$ in the structure containing the OCT-$8_{mer}$ in the presence of 5'-dAH+Met (Supplementary Fig. 3a). An alignment of the LIAS structure containing the OCT-$8_{mer}$ with that containing the 6S-OCT-$H_{pro}$ intermediate reveals a new conformation of the OCT moiety. In the 6S-OCT-$H_{pro}$ intermediate structure, the new bond between C6 of OCT and the sulfur from the auxiliary cluster forces C8 into a different orientation, 4.9 Å away from C5' of SAM (Fig. 5b and Supplementary Fig. 2a). However, no other significant changes in overall structures are observed (RMSD = 0.26 Å, Cα = 251 atoms).

SAM undergoes a reductive cleavage in the next step of catalysis to generate a second equivalent of the 5'-dA• and Met (Fig. 2, steps 4 and 5). This 5'-dA• abstracts a H• from C8 of the intermediate (Fig. 2, step 6), generating a C8-centered radical that attacks a second μ-sulfido ion of the auxiliary cluster, forming the lipoyl product (Fig. 2, steps 7 and 8). A snapshot of this step in catalysis is captured in the

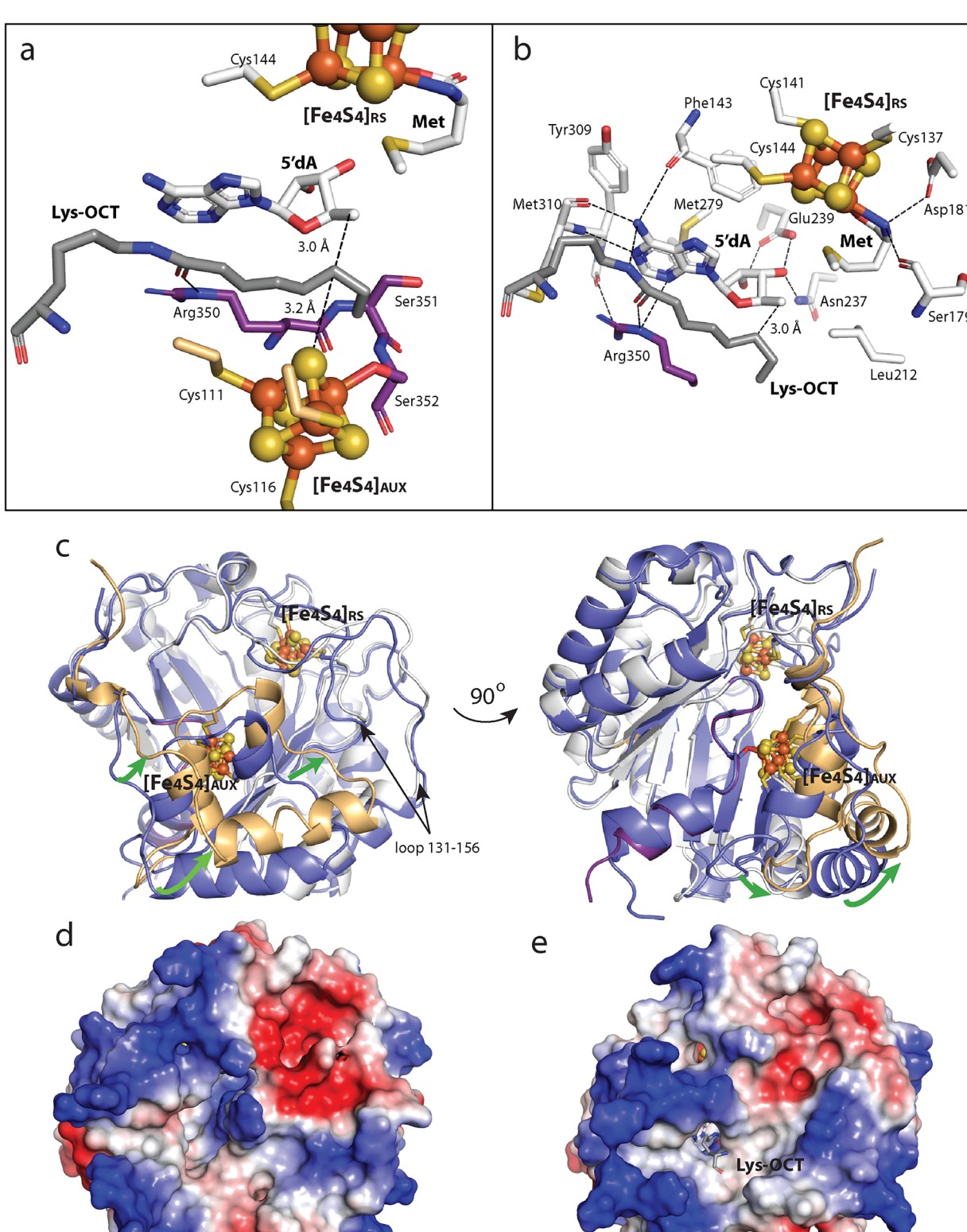

structure of the LIAS·6S-OCT-H_pro in the presence of 5′-dAH+Met, determined to 2.45 Å resolution (Fig. 5c and Supplementary Table 1). The active site of the complex shows the auxiliary cluster in the [Fe₃S₃:1 mercaptooctanoyl] state as in the LIAS·6S-OCT-H_pro structure in the presence of SAM. However, minor changes in the coordination between the 6S-OCT moiety and the adenine ring of SAM or 5′-dAH are observed. Because of these changes, SAM and 5′-dAH (the key players

in the active site) exhibit slightly different orientations, which result in changes in the distance between C5′ of SAM or 5′-dAH and C8 of the 6S-OCT-H_pro. The distance between C8 of the 6S-OCT-H_pro and C5′ of 5′-dAH is shorter (3.5 Å) in the structure with 5′-dAH+Met vs the comparable distance in the structure with SAM (4.9 Å) and is now perfectly positioned for H• abstraction from C8 (Fig. 5b, c and Supplementary Fig. 3b). The overall architecture of LIAS in complex with the 6S-OCT-

**Fig. 4 | Structure of human LIAS in the presence of the octanoylated 8$_{mer}$ peptide (OCT-8$_{mer}$) substrate.** Structure of **a** the LIAS substrate binding site in the presence of the OCT-8$_{mer}$ substrate containing an octanoyllysyl moiety (Lys-OCT) and 5′-deoxyadenosine (5′-dAH) +Met; **b** the SAM-binding site of LIAS in the presence of the OCT-8$_{mer}$ and 5′-dAH+Met. A series of conserved residues coordinate 5′-dAH and Met. The oxygen and nitrogen atoms of Met are coordinated to the unique iron of the RS iron-sulfur ([Fe$_4$S$_4$]$_{RS}$) cluster in a bidentate fashion. The amino group of Met forms H-bonds with the carboxyl group of Asp181 and the carbonyl oxygen of Ser179, while the C$_\gamma$ of Met is in hydrophobic contact with Leu212. The adenine ring of 5′-dAH interacts with the aromatic rings of Phe143 and Tyr309, while its C4 and C5 are in van der Waals distance from Met279. 5′-dAH forms H-bonds between its N1 and the amide nitrogen of Met310, its N3 and the N$_\epsilon$ of Arg350, its N6 and the carbonyl oxygens of Phe143 and Met310, and its ribose ring with both oxygens of the carboxyl group of Glu239 and the side chain nitrogen of Asn237. Cartoon representation of (**c**) superimposed structures of LIAS with no ligands (blue) and in the presence of the OCT-8$_{mer}$ and 5′-dAH+Met. LIAS color code: N-terminal domain (tan); RS domain (white); C-terminal α-helix (purple); iron–sulfur atoms of the auxiliary ([Fe$_4$S$_4$]$_{AUX}$) and [Fe$_4$S$_4$]$_{RS}$ clusters are represented as orange and yellow balls, respectively. The green arrows indicate conformational changes between the two structures. The electrostatic surface potential of LIAS with no ligands (**d**), and in the presence of the OCT-8$_{mer}$ and 5′-dAH+Met (**e**). Structures in **c**–**e** are in the same orientation.

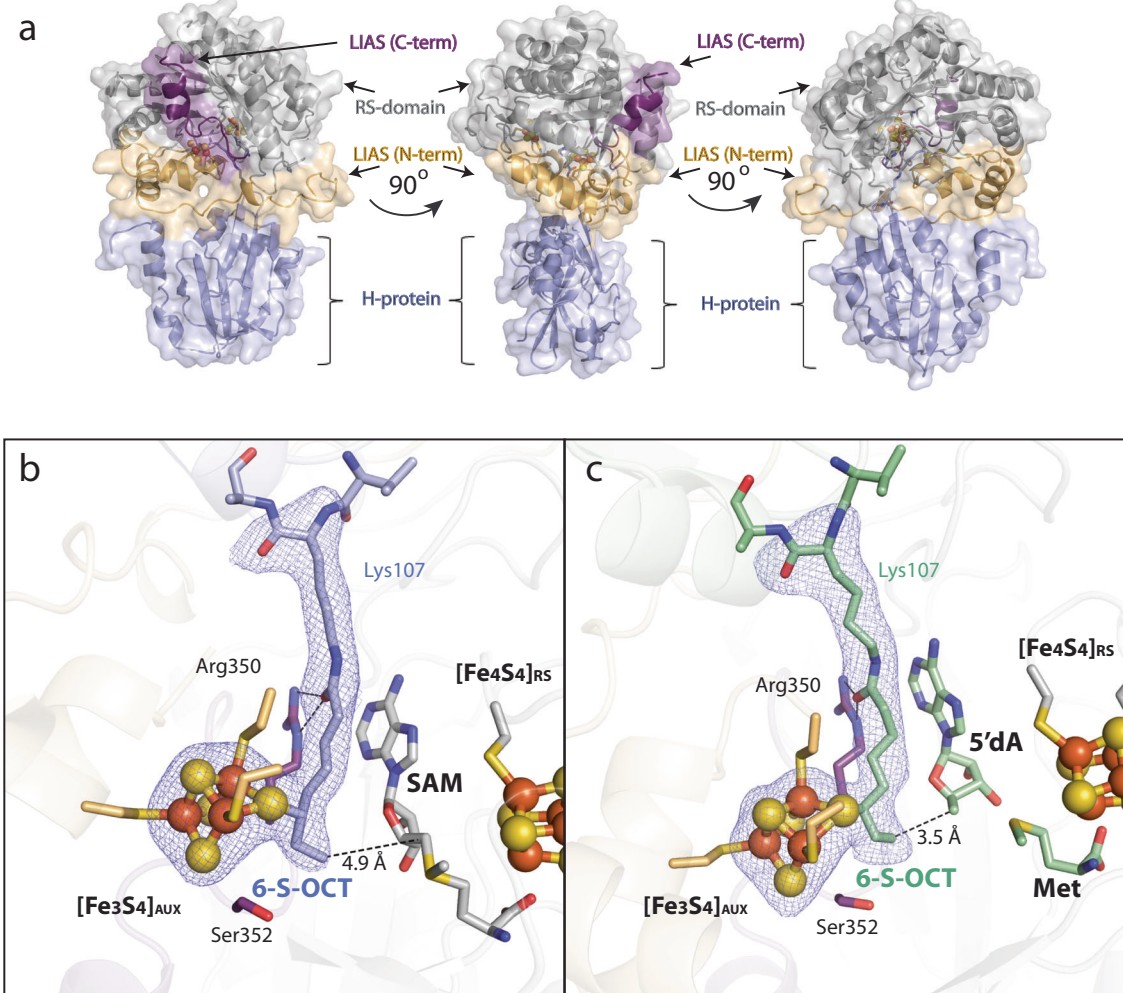

**Fig. 5 | A cartoon representation of the structure of the LIAS in complex with 6-mercaptooctanoyllysyl H-protein (6S-OCT-H$_{pro}$).** The overall structure of LIAS-6S-OCT-H$_{pro}$ in the presence of SAM (**a**); the active site of the LIAS-6S-OCT-H$_{pro}$ covalent complex in the presence of SAM (**b**) or in the presence of 5′-deoxyadenosine (5′-dAH)+Met (**c**). The mesh is contoured at 2.5 σ. LIAS color code: N-terminal domain (tan); RS domain (light gray); C-terminal α-helix (purple); iron–sulfur atoms of the auxiliary ([Fe$_4$S$_4$]$_{AUX}$) and radical SAM ([Fe$_4$S$_4$]$_{RS}$) clusters are represented as orange and yellow balls, respectively.

H$_{pro}$ and 5′-dAH+Met is highly similar to that with the OCT-8$_{mer}$ and 5′-dAH+Met (RMSD = 0.33 Å over 260 Cα atoms). However, a slight conformational change is observed in the loop containing residues 118–126 in structures containing SAM (Supplementary Fig. 3c and d). The changes in the positioning of the loop comprising residues 118–126 and the C-terminal helix in the structures containing 5′-dAH + Met show that the active site of LIAS is flexible and undergoes reorganization at different stages of catalysis (Supplementary Fig. 2b, c). Our structures do not provide concrete conclusions concerning which of the remaining sulfur atoms in the auxiliary cluster is

added to C8. Rotation around the C6-C7 bond of the octanoyl chain can result in two conformations with C8 at almost equivalent distances to the two closest sulfur atoms.

**Noncovalent interactions between human LIAS and H$_{pro}$**

The structure of the human H$_{pro}$ exhibits high similarity to that of the apo H$_{pro}$ from *E. coli* (PDB ID: 3A7L)[41], with an RMSD of 0.61 Å over 109 Cα atoms (Supplementary Fig. 4b). The H$_{pro}$ is composed of a central antiparallel β-sheet. This central β-sheet interacts with a peripheral β-sheet mainly through hydrophobic contacts and is surrounded by five

α-helices (Supplementary Fig. 4a). The conserved Lys107 is in the loop between β6 and β7 of the peripheral β-sheet and is modified with a 6-mercaptooctanoyl moiety. There are only two regions of non-covalent interactions in the structure of LIAS in complex with its H_pro substrate (Supplementary Figs. 5 and 6). The first region, encompassing residues 66–110, is in the N-terminal part of LIAS, while the second comprises residues 311–314 in a loop from the RS domain (Supplementary Figs. 5c, d and 6). Residues 66–110 form a belt that wraps around one side of the H_pro/LIAS interface. Several elements of the H_pro are involved in the interaction with LIAS through residues from β4, β6, β7, α-helices (1–3 and 5), and residues in the loops between β1and β2, β6 and β7, and α1 and β4 (Supplementary Figs. 5a, b and 6). The electrostatic surface potential at the interface between LIAS and the H_pro shows a network of charge-charge interactions. The H_pro exhibits mostly negative charges, counterbalanced by positive charges from LIAS (Supplementary Fig. 5a, c and 5b, d). Specific interactions between LIAS and the H_pro are shown in Supplementary Table 2 (Supplementary Figs. 5a, c, 6a, and 7a).

In *E. coli* and many other bacteria, LipA forms the LipCo directly on the octanoylated forms of all LCPs. In humans and many other organisms, especially eukaryotes, the LipCo is initially constructed only on the H_pro, and is then transferred to the other LCPs by LIPT1. This observation suggests that LIPT2 octanoylates only the H_pro, and perhaps LIAS only acts on the octanoylated H_pro. A comparative analysis of the human E2 subunits from the BCODC, KDC, and PDC reveals a high degree of similarity among them (Supplementary Fig. 7a). Superimposing the E2 subunits onto the structure of the H_pro in complex with LIAS shows the absence of the α1 and α5 helices in the E2 subunits, which are involved in complex formation with LIAS (Supplementary Fig. 7b). Furthermore, the E2 subunits display extended loops that clash with residues involved in substrate binding from the RS domain (Supplementary Fig. 7c). Our analysis suggests that these extended loops hinder the proper orientation of the OCT group in the active site of LIAS. This observation aligns with the finding that the E2 subunit of the PDC from *E. coli* lacks the extended loop (Supplementary Fig. 7d) and serves as a substrate for *E. coli* LipA following octanoylation by LipB[12]. Similarly, mammalian *LIAS* expression in *E. coli* complements a *lipA* knockout[42].

## Pathogenic variants of LIAS and H_pro in human health

Several mutations in the *LIAS* gene are associated with non-ketotic hyperglycemia (NKH) and directly affect lipoylation levels of the GCS and PDH complexes. For instance, a newborn carrying a homozygous mutation in the *LIAS* gene at c.475_477GAG > AAA, replacing the conserved Glu159 with Lys, was diagnosed with NKH three days after birth[43]. This child demonstrated a delay in mental development, loss of hearing, and seizures, and died before the age of three. Examination of the structure of LIAS in the presence of the OCT-8-mer substrate shows that the carboxyl group of Glu159 forms H-bonds with His192 and Met133. It is noteworthy that Met133 is near (8.1 Å) Cys137, one of the ligands of the $[Fe_4S_4]_{RS}$ cluster (Supplementary Fig. 8a).

The c.645 T > A mutation in the *LIAS* gene results in the substitution of Asp215 with Glu and was observed in a child exhibiting seizures, low muscle tone, and delayed mental development[43]. In the LIAS structure, the highly conserved Asp215 is in the loop of the RS domain near (6.7 Å) the $[Fe_4S_4]_{RS}$ cluster. It has alternating interacting partners in the LIAS structures at different catalytic stages. As discussed earlier, the changes induced by substrate binding lead to a decrease in the distance between the two clusters by -2 Å. The structure of LIAS without substrates (blue color Supplementary Fig. 8b) demonstrates that the carboxyl group of Asp215 forms H-bonds with the side chain of Gln256. Conversely, in the substrate-bound structure in the presence of 5dAH+Met, the carboxyl group of Asp215 forms an additional H-bond with a new partner, Arg139. Importantly, this conserved Arg139

is part of the $CX_3CX_2C$ $[Fe_4S_4]_{RS}$ cluster binding motif (Supplementary Fig. 8b).

Pathogenic mutations in both alleles of the *LIAS* gene, c.738-2 A > G from the mother and c.929 T > C from the father, have also been reported[44]. In this case, the child began developing symptoms at 19 months, exhibiting seizures and delayed mental development. One mutation leads to the substitution of Met310 with Thr, while the other results in improper splicing of LIAS isoforms 1 and 2. Met310 is highly conserved, although its peptide bond forms H-bonds with N1 and N6 of SAM or 5′-dAH. Of particular importance, the side chain of Met310 is in a van-der-Waals interaction with Lys107-OCT or Lys107-6-S-OCT of H_pro (Supplementary Fig. 8c). The Met310 to Thr substitution likely affects OCT-H_pro binding.

Other patients with neonatal-onset epilepsy, encephalomyopathy, muscular hypotonia, lactic acidosis, and defects in pyruvate and glycine metabolism were also reported to have genetic mutations in *LIAS*. These severe symptoms were linked to a homozygous single-nucleotide polymorphism c.746 G > A in the *LIAS* gene, resulting in the replacement of Arg249 with His[45]. Arg249 is a highly conserved residue that forms H-bonds with Asn237 and Val238 in the SAM-binding loop, as well as with Gln245 and Ser258, both of which are in two helices from the RS domain. In addition, Arg249 forms hydrophobic interactions with Gln245 and Glu239, where Glu239 is part of the SAM binding site (Supplementary Fig. 8d).

A recent study identified biallelic compound heterozygous variants in the *LIAS* gene: c.277delC (Lys92_Leu93insTer) and c.542 A > T (p.Asp181Val)[46]. The patient exhibited disorders such as developmental disabilities, epilepsy, microcephaly, strabismus, and chronic constipation starting at the age of 6 months. Biochemical analyses revealed slightly elevated glycine levels. The c.277delC variant was predicted to cause protein truncation, resulting in a complete loss of function. In contrast, the c.542 A > T (Asp181Val) variant was predicted to potentially alter catalytic activity by displacing the FeS clusters. Our structural study reveals that Asp181 is repositioned during conformational changes induced by substrate binding and forms an H-bond with the nitrogen atom of Met (a product of SAM cleavage) (Supplementary Fig. 8e).

The lipoylated H_pro, encoded by the *GCSH* gene, is crucial for two essential mitochondrial processes: the synthesis of the LipCo and its transfer to the E2 subunits of the BCODC, KDC, and PDC, as well as serving as a lipoyl carrier in the GCS. A study of six pathogenic variants in the GCSH gene demonstrated that most had a hypomorphic effect on both mitochondrial activities, leading to deficiencies in protein lipoylation and glycine metabolism, although some variants primarily affected only one of these functions. One example is the c.170 A > G mutation, which replaces His57 with Arg in the H_pro. The His57Arg H_pro variant demonstrated a minimal effect on glycine metabolism but significantly impacted the lipoylation of the E2 subunits of the mitochondrial complexes[47]. His57 in the H_pro is located near the LIAS interface in the complex (Supplementary Fig. 8f). Our computational analysis by MutaBind2 of LIAS's affinity to the His57Arg H_pro variant revealed a ΔΔG change of 1.55 kcal/mol, indicating a slight destabilization of the complex compared to that with the WT H_pro[48].

Other pathogenic mutations in the *GCSH* gene, including c.442 A > C (Thr148Pro) and c.344 C > T (Pro115Leu), exhibited hypomorphic effects on both mitochondrial activities of the H_pro. Thr148 and Pro115 are far from the interaction site with LIAS (Supplementary Fig. 8f); however, both variants were characterized by decreased levels of H_pro expression and, as a result, a significant reduction of lipoylated proteins.

## Discussion

LA is a critical cofactor in several multienzyme complexes important in primary metabolism. Many organisms have two pathways for

generating the LipCo: a de novo pathway and a scavenging pathway. Humans, however, only have the de novo pathway, which involves LIPT1, LIPT2, and LIAS. LIPT2 transfers the OCT chain from OCT-ACP to the $H_{pro}$. LIAS attaches two sulfurs, one to C6 and one to C8 of the OCT chain of the $H_{pro}$ to afford Lip-$H_{pro}$, while LIPT1 distributes the LipCo from Lip-$H_{pro}$ to the E2 subunits of the dehydrogenase complexes. Mutations in any of the genes encoding these proteins, as well as others involved in reconstituting LIAS's auxiliary cluster after each round of catalysis, are associated with mitochondrial energy metabolism disorders that lead to various health-related dysfunctions such as NKH, encephalopathy, epilepsy, and lactic acidosis[20,21,38,49–53]. These conditions are associated with delayed mental development, orthopedic problems, difficulties in feeding, and low muscle tone, and often cause early childhood death. In mice, homozygous embryos lacking LIAS die before 9.5 days post coitum[54]. Moreover, LIAS and LIAS-related proteins are predicted to play a significant role in some cancers through a newly described copper-dependent mechanism of cell death called cuproptosis[55–57].

Given the importance of LIAS in human health and disease, we determined the X-ray crystal structure of the protein. The structure resembles previously determined LS structures from *M. tuberculosis*[29] and *T. elongatus*[30], showing an open active site between two [Fe₄S₄] clusters. However, the *T. elongatus* LS structure had no biologically relevant ligand bound, while the *M. tuberculosis* LS structure only captured the stage of catalysis in which C6 of the OCT substrate is covalently bound to a partially degraded [Fe₃S₄] cluster. In this work, we determined LIAS structures during several stages of catalysis, including with a noncovalently bound peptide substrate and 5'-dAH+Met, which captures the stage of catalysis just before H• abstraction. We also determined the structure of LIAS in a complex with the entire $H_{pro}$ in the presence of SAM or 5'-dAH+Met. These two structures provide insight into the interaction between LIAS and $H_{pro}$, and capture stages in catalysis just after the formation of the crosslinked intermediate species and the binding of another SAM molecule, and just before H• abstraction at C8 to form the Lip-$H_{pro}$ product.

The active site of LIAS undergoes reorganization during catalysis, which occurs in two sequential half-reactions, enabling selective H• abstraction at each stage by a SAM-derived 5'-dA•. Our structural data indicate no changes in the positioning of the [Fe₄S₄]$_{RS}$ cluster or the 5'-dAH moiety of SAM in the substrate- or intermediate-bound structures. By contrast, significant changes in position are observed for the [Fe₄S₄]$_{AUX}$ cluster and the OCT moiety, which reduce the distance between the two Fe/S clusters from 13.7 Å to 11.8 Å in the substrate-free and substrate-bound structures, respectively. At the beginning of catalysis, the [Fe₄S₄]$_{AUX}$ cluster is coordinated by four ligands (Cys106, Cys111, Cys117, and Ser352) and is oriented such that only one of its sulfides is suitably positioned for attack by the C6-centered substrate radical. Attachment of sulfur at C6 occurs concurrently with the release of Ser352 and the loss of one Fe ion from the [Fe₄S₄]$_{AUX}$ cluster, resulting in changes in the active site that place C8 of the OCT group closer (~4.2 Å) to C5' of SAM. This distance decreases to 3.5 Å in the structure containing the [Fe₃S₃:1 mercaptooctanoyl] $H_{pro}$ intermediate with 5'-dAH+Met. Attachment of sulfur at C8 occurs only when the auxiliary cluster is in the [Fe₃S₃:1 mercaptooctanoyl] state, having lost coordination with Ser352 upon loss of the fourth Fe (Fig. 3c). The loss of the fourth Fe and the conformational rearrangement of Ser352 make the second sulfur accessible for insertion at C8. Much less is known about the immediate state of the auxiliary cluster after the formation of the lipoyl group. Our Mössbauer studies on the *E. coli* enzyme suggest that the cluster is fully degraded to ferrous ions in the absence of NfuA, which regenerates the auxiliary cluster after each turnover, although small amounts of Fe-containing species like [Fe₂S₂] clusters were also observed[31]. In the presence of NfuA, whatever remains in the auxiliary cluster site of *E. coli* LipA after a single turnover appears to be used for subsequent turnovers, given that one LipA protein can support the formation of two equivalents of lipoyl product[58]. Further studies to probe the detailed mechanism of auxiliary cluster regeneration are ongoing.

## Methods

### DNA and amino acid sequences
Below are the codon-optimized DNA and corresponding amino acid sequences of LIAS and the H protein used in this study.

**Truncated LIAS isoform 1 (65–372) optimized DNA sequence**
5'-AACCTAAAACGCCAGAAAGGAGAAAGGTTAA-GACTACCTCCATGGCTAAAGACAGAGATTCCCATGGGGAAAAATTA-CAATAAACTGAAAAATACTTTGCGGAATTTAAATCTCCATA-CAGTATGTGAGGAAGCTCGATGTCCCAATATTGGA-GAGTGTTGGGGAGGTGGAGAATATGCCACCGCCA-CAGCCACGATCATGTTGATGGGTGACACATGTACAAGAGGTTGCA-GATTTTGTTCTGTTAAGACTGCAA-GAAATCCTCCTCCACTGGATGCCAGTGAGCCCTACAA-TACTGCAAAGGCAATTGCAGAATGGGGTCTGGATTATGTTGTCCTGA-CATCTGTGGATCGAGATGATATGCCTGATGGGGGAGCTGAACA-CATTGCAAAGACCGTATCA-TATTTAAAGGAAAGGAATCCAAAAATCCTTGTGGAGTGTCT-TACTCCTGATTTTCGAGGTGATCTCAAAGCAATA-GAAAAAGTTGCTCTGTCAGGATTAGATGTGTATGCACATAATGTA-GAAACAGTCCCGGAATTACA-GAGTAAGGTTCGTGATCCTCGGGCCAATTTTGATCAGTCCCTACGTG-TACTGAAACATGCCAAGAAGGTTCAGCCTGATGTTATTTCTAAAA-CATCTATAATGTTGGGTTTAGGCGAGAATGATGAGCAAGTA-TATGCAACAATGAAAGCACTTCGTGAGGCAGATGTA-GACTGCTTGACTTTAGGACAATATATGCAGCCAA-CAAGGCGTCACCTTAAGGTTGAAGAATA-TATTACTCCTGAAAAATTCAAA-TACTGGGAAAAAGTAGGAAATGAACTTGGATTTCATTA-TACTGCAAGTGGCCCCTTTGGTGCGTTCTTCATA-TAAAGCAGGTGAATTTTTCCTGAAAAATCTAGTGGCTAAAAGAAAAA-CAAAAGACCTCTA-3'

**Truncated LIAS isoform 1 (65–372) amino acid sequence**
NLKRQKGERLRLPPWLKTEIPMGKNYNKLKNTLRNLNLHTVCEEARCPNI-GECWGGGEYATATATIMLMGDTCTRGCRFCSVKTARNPPPLDASE-PYNTAKAIAEWGLDYVVLTSVDRDDMPDGGAEHIAKTVSYLK-ERNPKILVECLTPDFRGDLK-AIEKVALSGLDVYAHNVETVPELQSKVRDPRANFDQSLRVL-KHAKKVQPDVISKTSIMLGLGEN-DEQVYATMKALREADVDCLTLGQYMQPTRRHLKVEEYITPEKFKY-WEKVGNELGFHYTASGPLVRSSYKAGEFFLKNLVAKRKTKDL

**H protein (a. a 49–173) optimized DNA sequence**
5'-AGCGTTCGTAAGTTCACCGAGAAACATGAGTGGGTTACCACA-GAAAACGGTATCGGTA-CAGTGGGTATCAGCAATTTTGCTCAGGAGGCATTAGGTGATGTTGTT-TACTGCTCTTTGCCGGAGGTCGGTACTAAATTGAACAAGCAA-GATGAGTTTG

CATTA-GAGTCTGTAAAAGCAGCGTCGGAGCTTTATTCCCCTTTGTCAGGT-GAAGTCACCGAAATTAACGAGGCTCTTGCTGAAAACCCAGGTTTGGT-TAATAAATCCTGTTACGAAGACGGCTGGTTAATCAA-GATGACCTTATCAAACCCTTCCGAACTTGACGAACTGATGTCTGAG-GAAGCATACGAGAAGTATATTAAATCCATCGAGGAA-3'

**H protein (a. a 49–173) optimized amino acid sequence**
SVRKFTEKHEWVTTENGIGTVGISNFA-QEALGDVVYCSLPEVGTKLNKQDEFGALESVKAASELYSPLSGEVTEINEA-LAENPGLVNKSCYEDGWLIKMTLSNPSELDELMSEEAYEKYIKSIEE

## Plasmids and strains

Genes encoding truncated human LIAS (aa 65–372) and human H protein ($H_{pro}$) lacking the mitochondrial targeting sequence (aa 49–173) were synthesized by ThermoFisher Scientific after they were codon-optimized for over-expression in *E. coli*. Utilizing *Nde*I and *Xho*I restriction sites, the *LIAS* gene was subcloned into pET-28a, while the gene encoding the $H_{pro}$ was subcloned into a modified pSUMO plasmid (LifeSensors Inc.), designated pDWSUMO, for protein over-production. After sequence verification by DNA sequencing at the Penn State Genomics Core Facility (University Park, PA), the resulting plasmids (pET-28a-trnLIAS and pDWSUMO-$H_{pro}$) were used to transform *E. coli* BL21 (DE3) competent cells. The cells used for the over-production of LIAS were also co-transformed with plasmid pDB1282, which harbors the *isc* operon from *Azotobacter vinelandii*[59]. As cloned, LIAS was expressed with an N-terminal $His_6$-tag separated from the protein by a ten aa linker, while the $H_{pro}$ was expressed as a fusion with a SUMO tag containing an N-terminal $His_6$-tag. During the purification of the $H_{pro}$, the SUMO tag was cleaved using Ulp1 protease, resulting in an $H_{pro}$ with an artificial Gly-His appendage at the N-terminus.

## Growth and expression of the LIAS and the apo H protein

LIAS and the $H_{pro}$ were overexpressed in *E. coli* using the following general procedure with the noted adjustments for the apo $H_{pro}$. A starter culture containing 25 µg/mL kanamycin (50 µg/mL for the $H_{pro}$) and 50 µg/mL ampicillin (ampicillin omitted for the $H_{pro}$) was inoculated with a single colony and incubated overnight at 37 °C with shaking at 250 rpm. A 25 mL aliquot of the overnight culture was used to inoculate 4 L of M9 minimal medium containing appropriate antibiotics as listed above and incubated at 37 °C with shaking (180 rpm) until an optical density at 600 nm ($OD_{600}$) of 0.3 was reached. For LIAS overexpression, the expression of the genes encoded on the pDB1282 plasmid was induced by adding 0.2% (*w/v*) L-arabinose at $OD_{600} = 0.3$. At $OD_{600} = 0.6$, 50 µM $FeCl_3$ and 100 µM L-cysteine were added before cooling the culture in ice water for 1 h with occasional shaking. For overexpression of the $H_{pro}$, L-arabinose, $FeCl_3$, or L-cysteine was not added. Protein expression was induced by adding IPTG to 0.2 mM, and the cultures were incubated further at 18 °C with shaking at 180 rpm for an additional 12 h. The cells were harvested at 4 °C by centrifuging at 6000×*g* for 12 min, flash-frozen in liquid nitrogen, and stored under liquid nitrogen until needed.

## Protein purification

The purification of LIAS was performed in an anaerobic chamber containing <1 ppm $O_2$ (Coy Laboratory Products, Grass Lake, Michigan), while the apo $H_{pro}$ was purified aerobically. Both proteins were purified by immobilized metal affinity chromatography (IMAC) using the following general procedures with minor modifications for the $H_{pro}$. Cells were resuspended in 200 mL of Lysis Buffer (100 mM Tris-HCl, pH 8.0, 150 mM KCl, 10 mM imidazole, 10 mM β-mercaptoethanol (BME), 10 mM $MgCl_2$). To the resuspended cells, the following were added at their indicated final concentrations: 0.25 mM $FeCl_3$, 1 mM L-cysteine, 1 mM pyridoxal 5′-phosphate (PLP), one SIGMAFAST protease inhibitor tablet (Sigma–Aldrich), 1 mM phenylmethylsulfonyl fluoride (PMSF), 0.5 mg/mL lysozyme, and 0.01 mg/mL DNase (no $FeCl_3$, L-cysteine, or PLP was added for the $H_{pro}$). The cells were disrupted by sonication with an ultrasonic cell disruptor (Branson Sonifier II "Model W-250", Heinemann), and the lysates were clarified by centrifuging at 4 °C for 1 h (45,000×*g*). The $His_6$-tagged proteins were then purified by IMAC using Ni-nitrilotriacetic acid (NTA) resin. The clarified lysate was loaded onto a Ni-NTA-containing column pre-equilibrated with Lysis Buffer. After loading, the column was washed with 150 mL of Wash Buffer (50 mM HEPES, pH 7.5, 300 mM KCl, 30 mM imidazole, 10% glycerol (*v/v*), and 10 mM BME). Then, the protein was eluted with 100 mL of Elution Buffer (50 mM HEPES, pH 7.5, 250 mM KCl, 300 mM imidazole, 10% glycerol, and 10 mM BME). The eluted $H_{pro}$ was

concentrated into a minimal volume and exchanged into Cleavage Buffer (50 mM HEPES, pH 7.5, 250 mM KCl, 5% glycerol, 40 mM imidazole, and 10 mM BME) using a PD-10 column (GE Healthcare). To remove the $His_6$-SUMO tag from the apo $H_{pro}$, ULP1 protease (50 µg per mg of protein to be cleaved) was added to the fusion protein, and the sample was incubated on ice overnight. The next day, the apo $H_{pro}$ sample was reloaded onto a Ni-NTA column pre-equilibrated in Cleavage Buffer. The protein was recovered in the flow-through, concentrated to a minimal volume, and exchanged into Storage Buffer (50 mM HEPES, pH 7.5, 250 mM KCl, 30% glycerol, 10 mM BME, and 2.5 mM Tris(2-carboxylethyl)phosphine (TCEP)) using a PD-10 column (GE Healthcare). For LIAS, the Ni-NTA-eluted protein was concentrated into a minimal volume and purified further by size-exclusion chromatography on a HiPrep 16/60 Sephacryl HR S-200 column (Cytiva). The S-200 column was connected to an AKTA protein liquid chromatography system (Cytiva) housed in an anaerobic chamber. The column was pre-equilibrated in Gel-filtration Buffer (50 mM HEPES, pH 7.5, 250 mM KCl, 10% glycerol, 10 mM BME, and 2.5 mM TCEP), and the protein was eluted at a flow rate of 0.5 mL/min[59]. Fractions containing the LIAS protein were identified by UV-vis absorption at 280 nm and were pooled, concentrated to a minimal volume, and exchanged into Storage Buffer.

The apo $H_{pro}$ was quantified by UV-vis absorption using a calculated extinction coefficient of 18450 $m^{-1}cm^{-1}$ at 280 nm (https://web.expasy.org/protparam/)[60]. In contrast, LIAS was quantified by the Bradford method using BSA (fraction V) as a standard and a correction factor of 1.6[61]. The purified proteins were aliquoted, flash-frozen in liquid $N_2$, and stored under liquid $N_2$ until needed. The purity of LIAS and the apo $H_{pro}$ was estimated by 12% sodium dodecyl sulfate-polyacrylamide gel electrophoresis (SDS-PAGE) and determined to be ≥95%.

## Synthesis of OCT-$H_{pro}$

The synthesis and purification of the octanoylated $H_{pro}$ followed a similar procedure as reported for *E. coli* $H_{pro}$, with minor alterations[14,62]. The reaction mixture contained the following in a final volume of 50 mL: 50 mM potassium phosphate buffer (KPB), pH 7.2, 15 mM $MgCl_2$, 5 mM ATP, 10 mM TCEP-HCl, 20 mM DTT, 5 mM octanoic acid, 200 µM apo $H_{pro}$, and 100 nM *E. coli* LplA. The reaction was allowed to incubate at 28 °C for 5 h, after which it was diluted 10× with 20 mM KPB, pH 7.2. The diluted protein was loaded onto a DE-52 column pre-equilibrated in the same buffer. The column was washed with 300 mL of 20 mM KPB, pH 7.2, containing 180 mM NaCl and then eluted with a 600 mL (total) linear gradient of 180–450 mM NaCl in 20 mM KPB, pH 7.2. The octanoylated $H_{pro}$ was concentrated to a minimal volume and buffer-exchanged into Storage Buffer (50 mM HEPES, pH 7.5, 250 mM KCl, 30% glycerol, 10 mM BME, and 2.5 mM TCEP) using a PD-10 gel-filtration column. The protein was aliquoted, snap-frozen in liquid $N_2$, and stored under liquid $N_2$ until needed.

## Protein crystallization

**The structure of human LIAS in the presence of DTT.** Crystallization of the $His_6$-tagged LIAS was carried out in a Coy anaerobic chamber. The protein solution was gel-filtered into the following buffer: 20 mM HEPES, pH 7.5, 0.2 M KCl, and 5 mM DTT. LIAS (8 mg/ml) was mixed with crystallization solution (0.1 M HEPES, pH 7.5, 0.2 M $Na_2SO_4$, 30% (*w/v*) PEG 3350, 3% (*w/v*) D-(+)-trehalose) in a 1:1 ratio. The hanging-drop vapor diffusion method was used against the same well solution to crystallize the protein, and the first brown crystals appeared after one week. The crystals were transferred into the following cryoprotectant solution: 50 mM HEPES, pH 7.5, 0.1 M $Na_2SO_4$, and 40% (*w/v*) PEG 3350. They were then mounted on the loops and frozen in liquid $N_2$.

Datasets were collected at the General Medical Sciences and Cancer Institutes Structural Biology Facility (GM/CA) or the Life

Sciences Collaborative Access Team (LS-CAT) beamlines at the Advanced Photon Source and the Howard Hughes Medical Institute protein crystallography beamlines (8.2.1 and 8.2.2) at the Advanced Light Source. The data images were processed using HKL2000[63]. Diffraction datasets for single anomalous dispersion (SAD) phasing experiments were collected at the Fe-absorption peak wavelength of 1.72 Å (Supplementary Table 1). The initial structure was solved using the SAD phasing method using the AutoSol program[64] and the automated protein model building program AutoBuild[65]. The higher-resolution native dataset was used for molecular replacement using the initial protein model coordinates and the Phaser program[66]. Coot[67,68] was employed to manually improve an initial model after molecular replacement and refinement using the Phenix.refine program[69].

**Structure of LIAS in the presence of 5'-dAH, Met, and the OCT-8$_{mer}$ peptide substrate.** LIAS crystals were obtained after co-crystallization of the protein in the presence of 5'-dAH, Met, and the OCT-8$_{mer}$ peptide. L-Met (2.5 mM) and 5'-dAH (1 mM) were sequentially added to the solution of LIAS (8 mg/ml) and incubated for 10 min at room temperature (RT). A 1.3× molar ratio of the OCT-8$_{mer}$ peptide was added to the protein and mixed with an equal volume of the crystallization solution described above. A similar solution was used as a cryoprotectant (50 mM HEPES, pH 7.5, 0.1 M Na$_2$SO$_4$, 40% ($w/v$) PEG 3350) except that 2.5 mM L-Met, 1 mM 5'-dAH, and 0.1 mM of the OCT-8$_{mer}$ peptide were added. The structure of LIAS with 5'-dAH+Met in the presence of the OCT-8$_{mer}$ substrate was solved by molecular replacement using the coordinates of LIAS in the presence of DTT as a starting model. The octanoyl-containing peptide substrate mimic, Glu-Ser-Val-($N^6$-octanoyl) Lys-Ala-Ala-Ser-Asp (OCT-8$_{mer}$), used in this study, was custom-synthesized by Proimmune (Oxford, UK).

**Structure of the LIAS–H$_{pro}$ complex in the presence of SAM or 5'-dAH plus Met.** A covalent complex between LIAS and the OCT-H$_{pro}$ was generated by incubating 450 µM LIAS and 450 µM octanoylated H$_{pro}$ in the presence of 450 µM SAM and 2 mM dithionite for 1 h at RT. The LIAS-6S-OCT–H$_{pro}$ covalent complex was purified by size-exclusion chromatography on a HiPrep 16/60 Sephacryl HR S-200 column (Cytiva) in a buffer containing 50 mM HEPES, pH 7.5, 0.2 M KCl, and 5 mM DTT and concentrated to 10 mg/ml. The purified complex was mixed with an equal volume of the crystallization solution (0.1 M K$_2$HPO$_4$ pH 8.0, 0.2 M KCl, 20% ($w/v$) PEG 3350) in the presence of either 1 mM SAM (Figs. 3b) or 1 mM 5'-dAH plus 2.5 mM Met (Fig. 3c). The crystals were transferred into a cryoprotectant solution (0.1 M K$_2$HPO$_4$, pH 8.0, 0.2 M KCl, 35% ($w/v$) PEG 3350) containing either 1 mM SAM or 2.5 mM Met and 1 mM 5'-dAH and mounted on loops and frozen in liquid N$_2$. The molecular replacement (MR) was used for phasing the X-ray diffraction dataset. Coordinates of LIAS for MR were obtained from a structure of the protein in the presence of 5'-dAH, Met, and the OCT-8$_{mer}$ peptide. Coordinates of H$_{pro}$ were obtained from an AlphaFold prediction. MR was performed using the Phaser program[66]. The initial modes were manually improved and refined using Coot[67,68] and Phenix.refine[69].

### Reporting summary
Further information on research design is available in the Nature Portfolio Reporting Summary linked to this article.

## Data availability
Atomic coordinates and structure factors for the reported crystal structures in this work have been deposited to the Protein Data Bank (PDB) under accession numbers 8TRW (LIAS); 8TSK (LIAS + OCT-8$_{mer}$ + 5'-dAH); 8UGO (LIAS-6S-OCT–H$_{pro}$ complex + 5'-dAH + Met), and 9C19 (LIAS-6S-OCT–H$_{pro}$ complex + SAM).

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

## Acknowledgements

This work was supported by the National Institutes of Health (awards GM-122595 to S.J.B.), the National Science Foundation (MCB-1716686), and the Eberly Family Distinguished Chair in Science (to S.J.B.). S.J.B. is an investigator of the Howard Hughes Medical Institute. This research used resources of the Advanced Photon Source, a U.S. Department of Energy (DOE) Office of Science User Facility operated for the DOE Office of Science by Argonne National Laboratory under Contract No. DE-AC02-06CH11357. Use of GM/CA@APS has been funded in whole or in part with Federal funds from the National Cancer Institute (ACB-12002) and the National Institute of General Medical Sciences (AGM-12006). The Eiger 16 M detector at GM/CA-XSD was funded by NIH grant S10 OD012289. Use of the LS-CAT Sector 21 was supported by the Michigan Economic Development Corporation and the Michigan Technology Tri-Corridor (grant 085P1000817). This research also used the resources of the Berkeley Center for Structural Biology, supported in part by the Howard Hughes Medical Institute. The Advanced Light Source is a Department of Energy Office of Science User Facility under contract no. DE-AC02-05CH11231. The ALS-ENABLE beamlines are supported in part by the National Institutes of Health, National Institute of General Medical Sciences, grant P30 GM124169. This research used 17-ID-1 Automated Macromolecular Crystallography (AMX) and 17-ID-2 the Frontier Microfocusing Macromolecular Crystallography (FMX) beamlines of NSLS-II, a national user facility at Brookhaven National Laboratory. Work performed at the CBMS is supported in part by the US Department of Energy, Office of Science, Office of Basic Energy Sciences Program under contract number DE-SC0012704. This article is subject to HHMI's Open Access to Publications policy. HHMI lab heads have previously granted a nonexclusive CC BY 4.0 license to the public and a sublicensable license to HHMI in their research articles. Pursuant to those licenses, the author-accepted manuscript of this article can be made freely available under a CC BY 4.0 license immediately upon publication.

## Author contributions

O.A.E. and S.J.B. developed the research plan and experimental strategy. O.A.E., D.M.W., and S.S.N. isolated proteins, and O.A.E. crystallized proteins and collected crystallographic data. O.A.E., D.M.W., and S.S.N. performed activity assays. O.A.E, and S.J.B. analyzed and interpreted crystallographic data. O.A.E., D.M.W., S.S.N., J.N.A., and S.J.B. wrote the manuscript, which was read and approved by all authors.

## Competing interests

The authors declare no competing interests.
