## [Transparent Peer Review File · Nature Communications]

Structural basis for catalysis by human lipoyl synthase

Corresponding Author: Professor Squire Booker

Version 0:

Reviewer comments:

Reviewer #1

(Remarks to the Author)

This ms essentially presents crystal structures. I am not a cryptographer and thus have only a few comments.

The human pathway of lipoic acid synthesis is nicely presented.

The demonstration of sulfur insertion at C8 of the octanoyl chain is a major step forward. The finding that reorganization of the chain relative to the auxiliary cluster and its dependence on C6 sulfur insertion and generation of the [Fe3-S3] cluster is clearly demonstrated and satisfying. This explains the obligate order of sulfur insertion in the bacterial LipA enzymes, although the archaea have a different mechanism that may be the primordial mechanism.

The interactions between GCSH and LIAS may provide insight into recently described clinical GCSH missense mutations (PMID 36190515). These mutations are rare relative to those in the other subunits of the glycine cleavage (glycine decarboxylase) system. This is presumably due to its much smaller target size.

Reviewer #2

(Remarks to the Author)

Manuscript NCOMMS-24-40825

In this manuscript, Dr. Esakova and her collaborators report various crystal structures of human lipoyl synthase (LIAS) in complex with different substrates, intermediates, and/or products to describe the different stages of lipoic acid biosynthesis. Lipoic acid is an essential cofactor for the synthesis of acetyl coenzyme A from pyruvate or various oxoacids, as well as in the glycine cleavage system. In humans, alterations in the activity of these enzymes or LIAS lead to several severe pathologies. LIAS belongs to the large family of radical SAM proteins that use the reduction of an FeS center to enable the cleavage of a SAM molecule and initiate a radical reaction. The distinctive feature of lipoyl synthase is that the sulfur atoms inserted come from sulfide ions derived from an auxiliary FeS center, which is destroyed during the reaction.

In their manuscript, the authors first report the structure of the enzyme without the SAM cofactor and octanoyl substrate and confirm that the human enzyme is very comparable to those of *M. tuberculosis* or *T. elongatus* previously determined. They then present two structures of the enzyme in complex with an octanoyl-peptide and either SAM or its cleavage products (5'-dA + methionine) and show that upon binding the substrate and SAM (or 5'-dA + methionine), the enzyme slightly closes to bring the two FeS centers closer together to facilitate the reaction. Finally, they report the structure obtained by incubating octanoyl-H-protein with LIAS and SAM under reducing conditions, leading to the formation of the lipoyl-H-protein adduct on one of the sulfur atoms of the partially degraded auxiliary FeS center into an Fe3S3-S-C6-octanoyl-H-protein center. This structure closely resembles the one already published in 2016 by some of the authors of this study, in which the same adduct was observed in the enzyme from *M. tuberculosis* (PDB ID 5EXK). The only notable difference in this new study is that the lipoic acid is linked to H-protein and not just to a peptide. This structure therefore helps describe the interactions between the two partners.

Overall, the data presented lack significant originality and merely confirm previously published results. For example, the fact that the formation of 5'-dA[•] enables either the C6 or C8 of the substrate to approach the C5' atom and thereby facilitate hydrogen atom abstraction has already been demonstrated in several other studies on radical SAM proteins. The structure of the adduct on the FeS cluster was also known, and ultimately, the presented work does not allow for the prediction of the second sulfide ion to be inserted at position 8. The interaction description between the H-protein and LIAS is very brief, with

no further analysis provided to explain the recognition specificity and why LIAS only recognizes the H-protein, unlike bacterial lipoyl synthases. Furthermore, the text contains several statements without actual experimental justification or explanation. To cite just two examples: In the last paragraph of the section "The overall structure of human LIAS," the authors claim that a shorter helix (containing the C117 residue) may provide further stabilization between the protein's N- and C-terminal regions, but do not explain how or provide any evidence for such further stabilization. Next, they assert that this shorter helix helps maintain an "open conformation" of the active site in the absence of the substrate, yet they again provide no experimental evidence to support this claim. For instance, do they mean that LIAS remains open while LipA does not? What is due to crystal packing and what is due to protein dynamics? These aspects are not discussed. Taking all these remarks into account, the reviewer believes that the data presented in this manuscript do not justify publication in a high-impact journal such as Nature Communications. Instead, the reviewer suggests that Biochemistry would be a more suitable alternative.

Minor points:

Beginning of the section: Thiolation at C6 of the substrate octanoyl moiety: the description of the mechanism has already been outlined in the introduction when discussing Figure 2.

Last paragraph of the introduction: "Also, Met was bound to the [Fe3S4]RS cluster, and the 5' carbon of 5'-deoxyadenosine (5'-dAH) was 4.2 Å from C6 and 3.7 Å from C8 of the octanoyl chain." [Fe3S4]RS should be [Fe4S4]RS

Reviewer #3

(Remarks to the Author)

Lipoyl synthase is a crucial enzyme responsible for the final step in the biosynthesis of lipoic acid, a vital cofactor required by several key enzyme complexes involved in energy production, amino acid metabolism, and fatty acid catabolism. By catalysing the insertion of sulphur atoms into the precursor molecule, lipoyl synthase enables the proper function of these complexes, which are essential for cellular respiration and metabolic homeostasis. Its importance is underscored by its evolutionary conservation across species and its role in preventing severe metabolic disorders linked to impaired mitochondrial function, making it indispensable for life.

LIAS is a rSAM enzyme that catalase the remarkable insertion of sulphur into the lipoyl cofactor at the expense of an auxiliary Fe-S cluster. The mechanism of LipA and the related chemistry in biotin synthase has been of interest for many years. The paper builds on previous structural studies to provide a much more complete and biologically relevant picture of the mechanism of catalysis. Using the authors novel approach, they were able to trap a number of crystallographic snapshots which have highlighted previously unknown structural rearrangements that occur.

The described experiments are technically sound, drawn conclusions are justified. Obtained results are of general interest to a broader readership. I recommend acceptance of the manuscript.

Version 1:

Reviewer comments:

Reviewer #1

(Remarks to the Author)

I had few concerns and they were suitably answered,

Reviewer #2

(Remarks to the Author)

In this revised version of their manuscript, the authors have addressed most of my comments. The new version is therefore acceptable without further modifications. Despite my previous remarks regarding the originality of this work, I would like to congratulate the authors on this impressive structural study, which, as they have emphasized, will undoubtedly have a significant impact on the scientific community.

Reviewer #1 (Remarks to the Author):

This ms essentially presents crystal structures. I am not a cryptographer and thus have only a few comments.

The human pathway of lipoic acid synthesis is nicely presented.

The demonstration of sulfur insertion at C8 of the octanoyl chain is a major step forward. The finding that reorganization of the chain relative to the auxiliary cluster and its dependence on C6 sulfur insertion and generation of the [Fe₃-S₃] cluster is clearly demonstrated and satisfying. This explains the obligate order of sulfur insertion in the bacterial LipA enzymes, although the archaea have a different mechanism that may be the primordial mechanism.

The interactions between GCSH and LIAS may provide insight into recently described clinical GCSH missense mutations (PMID 36190515). These mutations are rare relative to those in the other subunits of the glycine cleavage (glycine decarboxylase) system. This is presumably due to its much smaller target size.

We thank the reviewer for their positive comments and analysis of our work. In response to Reviewer 2's concern about the impact of our study, we mapped the mutations in the paper you cited onto the LIAS H_{pro} structure to gain insight into the roles of key disease-related mutations.

Reviewer #2 (Remarks to the Author):

Manuscript

NCOMMS-24-40825

In this manuscript, Dr. Esakova and her collaborators report various crystal structures of human lipoyl synthase (LIAS) in complex with different substrates, intermediates, and/or products to describe the different stages of lipoic acid biosynthesis. Lipoic acid is an essential cofactor for the synthesis of acetyl coenzyme A from pyruvate or various oxoacids, as well as in the glycine cleavage system. In humans, alterations in the activity of these enzymes or LIAS lead to several severe pathologies. LIAS belongs to the large family of radical SAM proteins that use the reduction of an FeS center to enable the cleavage of a SAM molecule and initiate a radical reaction. The distinctive feature of lipoyl synthase is that the sulfur atoms inserted come from sulfide ions derived from an auxiliary FeS center, which is destroyed during the reaction.

In their manuscript, the authors first report the structure of the enzyme without the SAM cofactor and octanoyl substrate and confirm that the human enzyme is very comparable to those of *M. tuberculosis* or *T. elongatus* previously determined. They then present two structures of the enzyme in complex with an octanoyl-peptide and either SAM or its cleavage products (5'dA + methionine) and show that upon binding the substrate and SAM (or 5'dA + methionine), the enzyme slightly closes to bring the two FeS centers closer together to facilitate the reaction. Finally, they report the structure obtained by incubating octanoyl – H-protein with LIAS and SAM under reducing conditions, leading to the formation of the lipoyl – H-protein adduct on one of the sulfur atoms of the partially degraded auxiliary FeS center into an Fe₃S₃-S-C6-octanoyl – H-protein center. This structure closely resembles the one already published in 2016 by some of the authors of this study, in which the same adduct was observed in the enzyme from *M. tuberculosis* (PDB ID 5EXK). The only notable difference in this new study is that the lipoic

acid is linked to H-protein and not just to a peptide. This structure therefore helps describe the interactions between the two partners.

Overall, the data presented lack significant originality and merely confirm previously published results. For example, the fact that the formation of 5'-dA• enables either the C6 or C8 of the substrate to approach the C5' atom and thereby facilitate hydrogen atom abstraction has already been demonstrated in several other studies on radical SAM proteins.

We agree that there are structures of RS proteins that show the 5'-dA carbon proximal to the site of hydrogen atom abstraction. In many of these instances, there is only one hydrogen atom abstraction in the reaction. In LIAS and a few other RS proteins, two molecules of SAM are used to generate two 5'-dA•s to regio- and stereoselectively abstract hydrogen atoms at C6 and C8. Before this work, how the enzyme chooses between C6 and C8 (and leaves C7 alone) at the beginning of the reaction was unknown because there wasn't a structure of the enzyme-substrate complex before catalysis. This is especially interesting because the substrate, being an aliphatic chain, makes no hydrogen bonds to the protein except at the amide carbonyl oxygen and nitrogen. One case in point is biotin synthase. The structure by Drennan and collaborators shows how the 5'-dA• can abstract the C9 hydrogen atom, but it is not clear how it would access the C6 hydrogen atom. Moreover, our previous structure, determined in collaboration with Drennan's lab, showed that the two clusters move closer together in the cross-linked intermediate. However, it wasn't clear whether substrate binding brings the clusters closer together or whether the formation of the intermediate does.

The structure of the adduct on the FeS cluster was also known, and ultimately, the presented work does not allow for the prediction of the second sulfide ion to be inserted at position 8.

We agree with the reviewer about our structures not allowing us to predict with high confidence which sulfide ions in the partially degraded cluster are added to C8. This line of research is currently under investigation. However, it's difficult because the remaining cluster is very unstable. We were able to generate two possibilities based on rotating around the C6-C7 bond of the octanoyl moiety. In one instance, we get about 2.8 Å away from the nearest sulfide of the auxiliary cluster, while we get 3.0 Å away in another. These distances are too similar to allow for an accurate prediction of what actually takes place during catalysis. This is a shortcoming of the work, and we state this in this section of the manuscript.

The interaction description between the H-protein and LIAS is very brief, with no further analysis provided to explain the recognition specificity and why LIAS only recognizes the H-protein, unlike bacterial lipoyl synthases.

We felt the initial description of the interaction between H-protein and LIAS was quite extensive and that more would have made the section too monotonous, especially given that all the interactions are described in Table S2. However, we performed some computational studies that might suggest why LIAS only recognizes the H-protein and not the E2 subunits of the dehydrogenase complexes. The mammalian E2 subunits have extended loops that would clash with part of the radical SAM domain of LIAS where the substrate binds. The mammalian H-protein does not have these loops. Our computational docking experiments using AlphaFold 3 show that H-protein docks well and delivers the octanoyl chain into the active site of LIAS. By contrast, the E2 subunits dock in a manner that places the octanoyl chain outside the active site of LIAS. It must be mentioned that there have been no in vitro experiments verifying that LIAS does not operate on E2 subunits. One likely possibility is that the specificity is determined by LipT2, which transfers the octanoyl chain from octanoyl-ACP to H_{pro}. All of this has been added to the manuscript.

Furthermore, the text contains several statements without actual experimental justification or explanation. To cite just two examples: In the last paragraph of the section "The overall structure of human LIAS," the authors claim that a shorter helix (containing the C117 residue) may provide further stabilization between the protein's N- and C-terminal regions, but do not explain how or provide any evidence for such further stabilization. Next, they assert

that this shorter helix helps maintain an "open conformation" of the active site in the absence of the substrate, yet they again provide no experimental evidence to support this claim. For instance, do they mean that LIAS remains open while LipA does not? What is due to crystal packing and what is due to protein dynamics? These aspects are not discussed.

We agree with the reviewer that those two statements are too speculative and have removed them. We have also looked through the manuscript to find other places where justifications are not provided and removed those statements.

Taking all these remarks into account, the reviewer believes that the data presented in this manuscript do not justify publication in a high-impact journal such as Nature Communications. Instead, the reviewer suggests that Biochemistry would be a more suitable alternative.

We appreciate the reviewer's thorough analysis of our work and respect their opinion. We want to point out that the lipoic acid biosynthesis pathway has many implications for human health and disease. Mutations in any of the genes encoding pathway proteins are associated with mitochondrial energy metabolism disorders that lead to various health-related dysfunctions such as non-ketotic hyperglycemia, encephalopathy, epilepsy, and lactic acidosis. Moreover, LIAS and LIAS-related proteins are predicted to play a significant role in some cancers through a newly described copper-dependent mechanism of cell death called cuproptosis. To add more impact, we have included a section that describes known disease-related mutations in LIAS and H_{pro} and where they map onto our LIAS/H_{pro} structure. Below is a synopsis of work on the human LIAS where our structural analyses can provide insight into lipoic acid-related pathways.

Work from Tracey Rouault's lab:

Jain, A., Singh, A., Maio, N., Rouault, T. A. (2020), Assembly of the [4Fe-4S] cluster of NFU1 requires the coordinated donation of two [2Fe-2S] clusters from the scaffold proteins, ISCU2 and ISCA1. *Hum. Mol. Genet.* 29:3165-3182

Crooks, D. R., Maio, N. Lane, A. N., Jarnik, M., Higashi, R. M., Haller, R. G., Yang, Y., Fan, T. W.-M., Linehan, W. M., Rouault, T. A. (2018), Acute loss of iron-sulfur clusters results in metabolic reprogramming and generation of lipid droplets in mammalian cells. *J. Biol. Chem.* 293:8297-8311

Work from Jim Cowan's lab:

Hendricks, A. L., Wachnowsky, C., Fries, B., Fidai, I., Cowan, J. A., (2021), Characterization and Reconstitution of Human Lipoyl Synthase (LIAS) Supports ISCA2 and ISCU as Primary Cluster Donors and an Ordered Mechanism of Cluster Assembly. *Int. J. Mol. Sci.* 22:1598

Work from Lucia Banci's lab:

Saudino, G., Ciofi-Baffoni, S., Banci, L. (2022) Protein-Interaction Affinity Gradient Drives [4Fe-4S] Cluster Insertion in Human Lipoyl Synthase. *J. Am. Chem. Soc.* 144:5713-5717

Camponeschi, F., Muzzioli, R., Ciofi-Baffoni, S., Piccioli, Banci, L. (2019) Paramagnetic ¹H NMR Spectroscopy to Investigate the Catalytic Mechanism of Radical S-Adenosylmethionine Enzymes. *J. Mol. Biol.* 431:4514-4522

Work from Roland Lill's lab:

Schulz, V., Basu, S., Freibert, S. A., Webert, H., Boss, L., Mühlenhoff, U., Pierrel, F., Essen, L. O., Warui, D. M., Booker, S. J., Stehling, O., Lill, R. (2023) Functional spectrum and specificity of mitochondrial ferredoxins FDX1 and FDX2. *Nat. Chem. Biol.* 10:205-207

Work from Peter Tsvetkov's lab

Bick, N. R., Dreishpoon, M. B., Perry, A., Rogachevskaya, A., Bottomley, S. S., Fleming M. D., Ducamp, S., Tsvetkov, P. J (2024) Engineered bacterial lipoyate protein ligase A (IplA) restores lipoylation in cell models of lipoylation deficiency. *J. Biol. Chem.* 300:107995

Dreishmpoon, M. B., Bick, N. R., Petrova, B., Warui, D. M., Cameron, A., Booker, S. J., Kanarek, N., Golub, T. R., Tsvetkov, P. J., (2023) FDX1 regulates cellular protein lipoylation through direct binding to LIAS. *J. Biol. Chem.* 299:105046

Tsvetkov, P., Coy, S., Petrova, B., Dreishpoon, M., Verma, A., Abdusamad, M., Rossen, J., Joesch-Cohen, L., Humeidi, R., Spangler, R. D., Eaton, J. K., Frenkel, E., Kocak, M., Corsello, S. M., Lutsenko, S., Kanarek, N., Santagata, S., Golub, T. R. (2022) Copper induces cell death by targeting lipoylated TCA cycle proteins. *Science*, 375:1254-1261

Work from Vamsy Mootha's lab

Joshi, P. R., Sadre, S., Guo, X. A., McCoy, J. G., Mootha, V. K. (2023) Lipoylation is dependent on the ferredoxin FDX1 and dispensable under hypoxia in human cells. *J. Biol. Chem.* 299:105075

Minor points:

Beginning of the section: Thiolation at C6 of the substrate octanoyl moiety: the description of the mechanism has already been outlined in the introduction when discussing Figure 2.

We agree with the reviewer but also appreciate that everyone might not follow the chemistry as well. We removed some of the language but wanted to retain enough to ensure the chemistry could be understood by a broad audience.

Last paragraph of the introduction: “Also, Met was bound to the [Fe3S4]RS cluster, and the 5' carbon of 5'-deoxyadenosine (5'-dAH) was 4.2 Å from C6 and 3.7 Å from C8 of the octanoyl chain.” [Fe3S4]RS should be [Fe4S4]RS

Thanks so much for catching this! It has been changed.

Reviewer #3 (Remarks to the Author):

Lipoyl synthase is a crucial enzyme responsible for the final step in the biosynthesis of lipoic acid, a vital cofactor required by several key enzyme complexes involved in energy production, amino acid metabolism, and fatty acid catabolism. By catalysing the insertion of sulphur atoms into the precursor molecule, lipoyl synthase enables the proper function of these complexes, which are essential for cellular respiration and metabolic homeostasis. Its importance is underscored by its evolutionary conservation across species and its role in preventing severe metabolic disorders linked to impaired mitochondrial function, making it indispensable for life.

LIAS is a rSAM enzyme that catalase the remarkable insertion of sulphur into the lipoyl cofactor at the expense of an auxiliary Fe-S cluster. The mechanism of LipA and the related chemistry in biotin synthase has been of interest for many years. The paper builds on previous structural studies to provide a much more complete and biologically relevant picture of the mechanism of catalysis. Using the authors novel approach, they were able to trap a number

of crystallographic snapshots which have highlighted previously unknown structural rearrangements that occur.

The described experiments are technically sound, drawn conclusions are justified. Obtained results are of general interest to a broader readership. I recommend acceptance of the manuscript.

We thank the reviewer for their positive comments and analysis of our work.